# A Critical Analysis of the Drivers of Human Migration Patterns in the Presence of Climate Change: A New Conceptual Model

**DOI:** 10.3390/ijerph17176036

**Published:** 2020-08-19

**Authors:** Rebecca Parrish, Tim Colbourn, Paolo Lauriola, Giovanni Leonardi, Shakoor Hajat, Ariana Zeka

**Affiliations:** 1Institute of Environment, Health and Societies, Brunel University London, Uxbridge UB8 3PH, UK; 2Institute for Global Health, University College London, 30 Guilford Street, London WC1N 1EH, UK; t.colbourn@ucl.ac.uk; 3Institute of Clinical Physiology, Italian National Research Council, 56124 Pisa, Italy; paolo.lauriola@gmail.com; 4Department of Public Health, Environments and Society; London School of Hygiene and Tropical Medicine, London WC1E 7HT, UK; Giovanni.leonardi@phe.gov.uk; 5Centre on Climate Change and Planetary Health, London School of Hygiene and Tropical Medicine, London WC1E 7HT, UK; Shakoor.Hajat@lshtm.ac.uk

**Keywords:** climate change adaptation, migration, climate migration, environmental migration, migration typology, global health, planetary health

## Abstract

Both climate change and migration present key concerns for global health progress. Despite this, a transparent method for identifying and understanding the relationship between climate change, migration and other contextual factors remains a knowledge gap. Existing conceptual models are useful in understanding the complexities of climate migration, but provide varying degrees of applicability to quantitative studies, resulting in non-homogenous transferability of knowledge in this important area. This paper attempts to provide a critical review of climate migration literature, as well as presenting a new conceptual model for the identification of the drivers of migration in the context of climate change. It focuses on the interactions and the dynamics of drivers over time, space and society. Through systematic, pan-disciplinary and homogenous application of theory to different geographical contexts, we aim to improve understanding of the impacts of climate change on migration. A brief case study of Malawi is provided to demonstrate how this global conceptual model can be applied into local contextual scenarios. In doing so, we hope to provide insights that help in the more homogenous applications of conceptual frameworks for this area and more generally.

## 1. Introduction

The climate change–migration nexus has been the subject of research debate for decades. Indeed climate change has been instigated in human migration since early humans first moved out of Africa and migration has long been an adaptive strategy to climate shocks, long-term changes or cyclic climate conditions [1]. The field of climate migration has been gaining global scientific and popular attention since roughly the 1970s [2], and very much so in recent years, since the emergence of the concept of ‘environmental refugees’ [3]. Since the 2015 European migration ‘crisis’, the topic has received increasing controversy and non-evidence-based rhetoric in the media. As climate change continues throughout the 21st century, it will likely serve as a threat magnifier of other migration drivers [4].

Whilst terms such as ‘climate refugee’ are not recognised legally, migration and conflict are considered key mechanisms by which climate change has become a priority global health concern [5,6,7]. Indirect health impacts of climate change, such as those mediated via migration and displacement, are often under-recognised and under-researched. The ongoing Covid-19 pandemic is a poignant example of how the special circumstances of migrant communities creates unique and extreme health vulnerabilities: whilst some migrants living in displacement camps are unable to practise good hygiene and social distancing, other migrants are finding themselves denied their right to asylum, neglected or turned away at borders due to travel restrictions and fear of new waves of infection. The Lancet Countdown on climate change and health created a ‘climate migration’ indicator [4] whilst the newly launched Lancet Migration collaboration aims to explore and provide evidence for policy on the impacts of climate change on migrant health [8].

Despite these advances, conceptual frameworks for robustly exploring and understanding the impacts of climate change upon migration are lacking in some key areas, and the body of empirical studies remains thin. These gaps undermine the ability of policy makers to design effective evidence-based policy, public health interventions, and strategies to support safe and positive migration experiences.

The aims of this paper are to provide a critical review of existing climate migration literature; from this, we also suggest modifications to existing conceptualisation of climate migration by providing a new conceptual model of the system of migration determinants. The model is designed to be pan-disciplinary and transferable to any geographic or social context. We advocate the systematic application of theory in climate migration studies which may help better geographic representation [9] and improve our understanding of how climate change is impacting migration with contextual relevance to policy makers and public health interventions. Finally, we apply this model to a case study of Malawi to demonstrate how doing so can improve understanding of the local context and result in well-grounded and policy-relevant insights into the true impacts of climate change on migration.

## 2. A Critical Review of Climate Migration Literature

In order to improve conceptual modelling, several critical issues related to climate change and migration have been identified and discussed here. A key characteristic of migration is the multicausal nature of its drivers. Climate change may act as a direct driver of displacement but in many cases is also inextricably linked to other, dynamic and interacting social, political, demographic and economic drivers [10,11,12]. A popular constructed narrative is that climate change acts as a threat magnifier of existing migration drivers. This can result in many empirical studies identifying that economic factors rather than climate factors dominate the decision to migrate [13,14,15]. However, it is possible that such studies overlook the mediated effect of climate change through other factors such as agriculture [16]. It is now largely acknowledged that the relationship between climate change and migration is complex, dynamic and non-directional.

Another critical issue relates to the multifaceted nature of climate change itself. Scholars typically outline several classes of climate change: a change in climate variability; changes in frequency and magnitude of fast-onset climatic events (including extreme weather events, droughts, floods, and heatwaves); and slow-onset climate change, including long term changes in average temperature, rainfall and chronic drought or flooding [17]. This wide temporality as well as severity of climate change must be accounted for when discussing the implications of climate change on future population movements.

Migration itself may occur over a range of spatial scales—from movements between rural and urban areas, to international migration—as well as a range of temporal scales such as short-term migration, circular migration, to permanent moves. The decision of each individual to migrate may also carry different levels of human agency. The decision to migrate is due to an aggregation of micro-level (typically household or individual) and macro-level (societal) drivers. As such, each potential migrant has their own unique profile of factors and drivers. Such individualistic situations are often described in terms of the individual’s or community’s vulnerability [18,19,20]. This presents a key challenge in many existing studies, which struggle to reconcile drivers at the macro- and micro-demographic scales.

A key challenge remains the paucity and compatibility of datasets regarding both migration and potential drivers thereof, and the scarcity of such data at appropriate spatial and temporal levels, particularly in low-income and in indigenous communities. The necessity for localised quantitative studies can result in fragmented analyses of specific timescales, geographies, types of migration and drivers thereof. Furthermore, this can make it difficult to summarise and build a global narrative of the risks of climate change to human security [21].

The resultant synthesis is that the impacts of climate change on migration are complex, multifaceted and dynamic. As such, increased attention on the upstream drivers of migration is called for.

Some authors argue that there is also some limitation in theoretical development and so in recent years there has been a push to promote a more sophisticated theoretical understanding of how climate change may interact with other drivers of migration [11,16,22]. This has allowed the narrative to evolve through time: the conventional narrative suggests a more simplistic view of climate change as a blanket push driver resulting in large-scale waves of migration [18]. However, newer frameworks appreciate the multi-driver nature of migration as well as the resilience and adaptive strategies of affected individuals and communities. Nevertheless, such frameworks still struggle to capture the dynamic nature of such drivers including feedback and lag times, as well as the interactions between drivers themselves through time. The rising concept of ecological public health goes some way to attempt to address this, yet many policy-facing groups remain slow on the uptake [23]. Understanding of climate-induced migration is further skewed by the discipline of researchers: each scientific discipline—be it epidemiology, economics, political sciences or anthropology—carries with it its own intrinsic assumptions and methodologies [22,24,25] which can perpetuate the fractured nature of the literature. Therefore, to advance the understanding of the impacts of climate change on migration requires a truly interdisciplinary response.

The collective result of these challenges is that current understanding of climate-induced migration is geographically uneven and studies are often non-transferable to other settings which presents an obstacle for policy makers and health intervention design. To aid design of interventions, scientists and decision makers (such as governments and humanitarian agencies) should engage with each other at all points of the intervention design and implementation. This can help ensure that interventions are contextually relevant and evidence based and that their impacts can be measured and evaluated.

## 3. A New Conceptual Framework

### 3.1. Migration Typology

From the challenges identified in the above critical analysis, a new conceptual framework of climate migration is provided. Migration, as a subjective concept, cannot carry one single definition and is highly contextualised. However, this is often neglected within most quantitative studies. In particular, the terms ‘environmental migration’ and ‘climate migration’ lack unanimous definitions across academic, NGO and political actors. Migration exists as a normative behaviour in most communities globally but may also manifest as forced displacement and other involuntary or voluntary movements. Furthermore, often overlooked in climate migration literature is the possible inhibiting effect of climate change on migration, resulting in reduced mobility rather than driving migration events [26,27].

Many scholars advocate the need for appropriate migration typologies and several have been presented. For instance, Stojanov et al. [2] argue the need for contextualisation of climate drivers on a community in order to appropriately discern climate driven migration from normative or otherwise induced movement. Renaud et al. [28] also comment on the difficulty of identifying the environmental signal within migration drivers and present a decision-making framework and accompanying typology of environmentally induced migration. Carling [29] created the first iteration of the aspiration-capability framework, which describes voluntary or involuntary, mobility or immobility, based on both desire and ability to migrate.

Based on a review of a large variety of both qualitative and quantitative literature, we identify four dimensions which quantify migration: societal, temporal, spatial, and agency levels. “Societal level” refers to the level of society affected, from micro scale (individual and household level) to macro scale (community, regional or population level). “Temporal level” refers to the time duration of the migration, and the short term may consist of a matter of months, the long term is typically considered to be a year or more, though there is much range within empirical studies, and permanent migration represents the longest form of migration. “Spatial level” refers to the physical distance covered by the migration. Short distance may consider anywhere from intracommunity, intra-regional movements, to movements within the country and includes movements to between rural and urban hubs. Long distance constitutes international movements across large geographical areas. Whilst some cross-border movements may only require a few miles of travel and as such may be considered short distance, a large amount of international movements cover multiple countries and sometimes continents. Such movements are of international political interest, though do not represent a large quantity of migrants or types of mobility [30,31]. The spatial scale, like the societal scale may also be summarised in terms of macro (generally medium or large distance) and micro (small, community level distances) and may align to climate and economic macro- or micro-level determinants. “Agency level” refers to the level of choice afforded to each migrant, existing as a continuous scale between the extremes of totally involuntary (in other words, forced) to totally voluntary. It should be noted that all four dimensions are continuous variables and hence demarcations used should be contextually modulated. By applying generalised demarcations, however, we classify five key categories of environmentally induced migration. The first category is forced displacement, also referred to as distress migrants [20] or temporary displaced migrants [2]. The second category is adaptive migration at the decision of the migrant(s) [11]. Whilst this is a voluntary movement, a crucial caveat is applied here to note that such migration may not be truly voluntary; whilst many scholars and decision makers consider it as such, there is an emergent narrative arguing that migration due to longer-term environmental or economic degradation, or erosion of human security constitutes a type of forced migration, rather than a voluntary or adaptive movement [32]. The third category is proactive migration at the decision of wider authority such as local or national government referred to as ‘planned resettlement’ [5]. The fourth category is for trapped populations, which refers to a lack of mobility due to at-risk populations becoming trapped by environmental and socioeconomic barriers such as poverty [32]. The final category is immobility, which represents a lack of mobility at the decision of the person(s) at environmental risk [32,33].

Table 1 summarises these classifications according to the four dimensions outlined. Throughout the remainder of this paper, we shall use the term ‘climate migration’ for simplicity.

### 3.2. Drivers of Migration

A newly proposed conceptual framework for identifying the determinants of any given migration is presented in Figure 1 below. This framework is an updated iteration of prior models, which, over time, have converged to agree that migration is generally the result of a combination of upstream drivers, split into five categories: social, economic, political, demographic and environmental. However, consideration of interactions and evolution of these drivers through time, societal and spatial scales remains low. Climate change is presented as an external driver which is expected in many contexts to act as a threat magnifier by exaggerating negative, push factors for vulnerable populations [34,35,36]. Attributes of climate change are split into three categories: physical, biological/ecological, and anthropogenic impacts [37,38]. The physical effects of climate change may be fast onset or slow onset. Fast onset includes sudden events such as extreme weather or disaster events. Slow onset consists of more gradual changes of mean values such as annual rainfall, rainfall variability and chronic droughting and flooding. Secondary, or ecological climate aspects may include changes in land cover, flora and fauna habitats, including disease vectors and pollinators. Tertiary or anthropogenic aspects include subsequent changes to anthropogenic systems such as crop yield and fish or game catch.

The model aims to build upon this a priori understanding by providing deeper discussion of the complexity of driver interactions, driver dynamicity and the evolution of both drivers, and their linkages over time and spatial scales. Importantly, the range of possible migration outcomes receives greater attention in this conceptual model, with recognition that different combinations of causal and contextual determinants may result in different migratory responses, including differences in the level of agency of a migrant. The model is designed with the purpose of being pan-disciplinary and, as such, relevant in any academic or non-academic context. The model presents not only a theoretical exercise, but a frame of thinking to support quantitative studies, with a view to informing future research, data collection or intervention design.

Drivers within each of the five classes may act as push agents—encouraging movement away from the origin, or pull agents—attracting movement to a host area. Bowles et al. [38] also identify glue and fend factors. Glue factors act to cement a potential migrant in his/her home location such as cultural and family ties, whilst fend factors deter migration into an area such as hostile immigration policies. Recent studies highlight that climate change may act as a glue factor in many situations, rather than as a push factor as is often popularised [26]. Climate change is a sub-category of environmental drivers which may be further categorised into three classes [10,37,39]. These are perhaps best described as primary physical effects, secondary biological and ecological effects, and tertiary anthropogenic effects [37,38]. Climate change is segregated here from other environmental factors and framed as an externality to the determinant system. This facilitates investigation of the impacts of climate change as an upstream pressure to all five classes of drivers. Each of the five categories is described in further detail below.

To demonstrate the temporal nature of the system, the model is presented on a set of axes with time on the horizontal dimension with arbitrary timepoints t_0_ and t_1_. This encourages consideration of the dynamicity of all determinants, as well as the changing nature of their interactions through time. As such, feedback implication on both host and source environments and communities of a migration decision may be decoded. Externalities, such as future climate shocks or political interventions, such as climate mitigation, which may alter the system and resultant migration can also be presented and their impacts conjectured. The *y* axis depicting scale of impact refers simultaneously to the societal and spatial level of impact thereby encouraging the disparate nature of drivers on these scales to be considered. Micro refers to small-scale, individual- or household-level factors whilst macro may be factors affecting large distances and large numbers of people.

Within the next section, we take a more granular view of each of the key families of drivers, and consider how each may directly or indirectly impact migration. This analysis is not exhaustive but attempts to provide a detailed summary of drivers over time and space, thereby encouraging a more nuanced and detailed exploration of the complexity of climate change.

#### 3.2.1. Climate Change

All climate factors are considered to occur at the macro spatial scale (Figure 1), which correlates to the macro societal impact level as described in Table 1 above. Temporality of climate factors varies, and depends on the climate determinant (as shown in Table 2). The true speed of climate change varies geographically and so there can be no definitive definition for fast or slow onset. Furthermore, some aspects may manifest across multiple timeframes.

##### Physical Effects

It is well established that climate change is increasing the frequency and magnitude of extreme weather events and climate shocks, which can be a direct cause of forced displacement. Nevertheless, in such natural hazard events, socially constructed vulnerabilities often govern the extent and type of migration responses which occur. Myers et al. [40], in a study of displacement due to Hurricane Katrina in 2005, identified a range of social vulnerability factors which had a significant impact upon outmigration from affected places. Similarly, Gray and Bilsborrow [14] identified within an Ecuadorian household migration survey, that household vulnerability factors such as home ownership, connectedness of household (to roads and schools), and poverty level, all confounded the environmental signal in the causes of observed migrations.

Less clear is the extent to which long-term or chronic climate change affects migration. Chronic changes may include changes in average temperature, average rainfall, rainfall variability or extent of periodic drought and flooding. Such changes often impact migration via mediating biological and anthropogenic factors such as impeded agricultural outputs [13,41], adverse health outcomes [42,43], or labour productivity [44]. In such examples, the extent of climate factor as a driver of migration compared to other sociodemographic and economic factors is seen to vary greatly across studies. To better understand such relationships, we classify these indirect impacts as biological or anthropogenic (secondary or tertiary).

##### Biological/Ecological and Anthropogenic Effects

Biological or secondary impacts are as a result of physical climate change, which may lead to changes in regional geochemistry, and flora and fauna. Such biological changes may alter the vulnerability of human populations. For instance, climate change may drive changes in the distribution of disease vectors [37,45].

Anthropogenic or tertiary aspects of climate change comprise the resultant alterations to human systems. Examples may include changes in anthropogenic land use and land availability due to sea level rise. Alterations, for example, in crop yield and fish catch, may have direct implications to socioeconomic factors, for example, due to reduced agricultural output [41], food security [1], and therefore upon urbanisation rates due to rural to urban migration [38]. Such anthropogenic pathways generally act over a longer temporal scale and can lead to the climate signal being masked by more proximal factors. As such, understanding of their impact on migration remains inconclusive [46] and less studied than direct physical impacts [39]. Furthermore, additional consideration is needed to assist the recognition of dynamic interactions between the physical, ecological and anthropogenic aspects of climate change.

#### 3.2.2. Other Migration Drivers

There are of course a range of other drivers of migration which are important to understand as well as how they may be affected by climate change. It is usually a combination of drivers that culminates in an individual’s decision to migrate and in what manner. Within the context of climate change, we refer to this aggregation of drivers (climatic and other) as the ‘vulnerability profile’ which will be unique to each individual. We now present a more detailed view of some key drivers within each of the five main classes identified. These are outlined in Table 3 below.

As well as existing as intermediary drivers, each of these drivers may have direct impacts on migration decisions. For example, Henry et al. [13] using regression modelling and Gray and Bilsborrow [14] using discrete time event history modelling both found that high literacy rates and economic status can act as significant push factors for migration. Ezra and Kiros [15] also found marital status and poverty level acted as push factors. Warner et al. [1] clearly identified the role of government relocation policy on driving planned resettlement of communities away from flood plains in Mozambique. Such epidemiological methods are well utilised for analysing such direct causes. However, each analysis is limited to a specific type of migration and set of pre-assumed key drivers. It is not possible within this paper to examine in depth the nature and relationships of each driver, rather the authors focus on presenting a broad overview, elucidating the multilevel and multitemporal nature of migration drivers, as well the dynamicity of the drivers and their linkages. Some key examples are used to demonstrate such complexities.

##### Social Drivers

Many studies examine the importance of social factors such as migration networks [47,48,49,50]. Education and literacy rates have also commonly been identified as determinants of vulnerability [35,51,52,53,54].

Social drivers such as education and poverty may also alter other drivers. For example, other studies have identified that in poor areas of Malawi where rain-fed agricultural practices reigned, the predominant climate change adaptation approach was not seasonal migration but the introduction of irrigation techniques to increase crop yields [54,55]. However, Joshua et al. [55] concluded that increased irrigation triggered increased water insecurity and hence water conflict. This interaction between poverty and adaptation approach has significant implications for future vulnerability levels and on future social and political factors. Of course, such impacts are not isolated to only impoverished communities. Developed countries with lower poverty levels can also suffer compound impacts of climate change on other social determinants [56]. However, developed countries generally have a higher capacity to mitigate or adapt to such changes resulting in different outcomes (migration and other), with different distributions across communities [57].

##### Economic Drivers

Closely linked to social factors are economic considerations such as employment opportunity and household wages [58] at the micro societal level. Poverty is also a key determinant of an individual’s vulnerability and hence ability to migrate [1,14,15]. Macro-level factors such as average employment rates and average income of a community may also act as push or pull factors which have often been identified as the dominant drivers of migration [11,41]. There also exists a debate on the role of failed politicised economic models such as ‘trickle down’ and ‘rent seeking ‘as being largely responsible for the increase in wealth gaps and rising relative poverty [22]. These economic models may contribute to future migration behaviour due to relations between poverty and mobility [1] and the effect of inequality gradients acting as sinks for migration [59]. For instance in the Malawian example given above, Findlay [54] also comments on the additional causes of food insecurity beyond water scarcity, including soil erosion, socioeconomic factors including vulnerability to poverty, ability to financially withstand crop failures, low food utilisation and infrastructural factors such as high transport costs.

##### Political Drivers

Political drivers are largely absent from environmental migration quantitative studies and yet present a significant category of migration drivers. Possibly the most influential and most studied of this category is the role of political insecurity on migration. Whilst the role of political insecurity and conflict is a well-acknowledged driver of migration, the role of climate change in driving political instability remains contested [20]. Burrows and Kinney [6] present an overview of multiple pathways through which climate change may lead to or exacerbate conflict such as through increasing rural to urban migration, resource competition or dispute between migrant and host communities. Sokolowski et al. [60,61] also discuss the role of political interventions such as efforts for conflict resolution, international relief, and immigration policy such as the closing of borders, and their impact upon migration outcomes.

Though largely overlooked in general climate migration literature, some models do focus on political drivers of migration with relatively accurate predictions [59,62]. Sokolowski and Banks [60] modelled population displacements that occurred in Syria in 2013 using UNHCR guidelines for factors prompting departure. Indeed, the Syrian conflict can be argued to contain both political and climate determinants in the mass displacement that has resulted [63].

Other political drivers include level of governance and trust in government and the level of institutionalisation and infrastructure within a community. Infrastructure and governmental and non-governmental organisations are critical intervention nodes and as such their connection to environmental migration form an important area of potential study. Other policies such as water, food and agricultural policy also co-interact and may result in a range of normative and adaptive migration approaches. For example, Loevinsohn [64] studied the 2002 Malawian food crisis and identified primary causal factors to be both environmental drought and the underinvestment by the national government in agricultural stock. Loevinsohn further identified that 39% of households interviewed during 2002 had migrant family members working seeking alternative income [64].

Crackdown on immigration policies in Western countries such as Britain, the USA and across the EU will also have significant impact upon future migration trends. With climate change expected to impact the numbers of both internal and international migrants in the future, existing dichotomies between the evidence on migration drivers and the political response to it will undoubtedly renew pressure on migration issues [65].

##### Demographic Drivers

Demographic factors at the micro level (such as age, gender, ethnicity) as well as at the macro level (such as average living conditions, affluency, diaspora presence) can act as push or pull factors as well as interact with other factors. The combined effect of climatic drivers and demographic drivers have resulted in many developing countries being the most vulnerable nations to climate change and has helped to drive research and narratives around climate justice [66] and climate refugees [67]. Rapid urbanisation is often a trend in such locations, leading to slum development, poor infrastructure and high vulnerability to future climate change, not to mention other shocks such as the Covid-19 pandemic.

In developed countries, different demographic challenges such as population ageing may also impact upon population mobility and health. For example, an older population may result in a reduced willingness to move and increased mental health burden of doing so [68]. Conversely, countries with aging populations can benefit from the ‘healthy-migrant’ effect [69]. As such, appreciating the demographic factors, their dynamics and interactions is essential to understanding climate risk on future sustainable development and population changes. When modelling future environmental migration, it is therefore essential to take into account the demographic situation of the study area.

##### Environmental Drivers

Climate change is a key driver of environmental change. Environmental degradation, such as desertification, permafrost melt and coastal erosion, undermines livelihoods and therefore acts as a push driver for migration away from these regions. In the short term, there may be positive environmental changes such as increased precipitation and improved agricultural production in many parts of the globe which may act as a migration pull factor [58]. Environmental determinants such as rainfall and vegetation cover are commonly used in quantitative studies of climate change though other ecosystem attributes and ecosystem degradation appears somewhat overlooked in migration studies, such as food availability from natural sources and pollution of water.

Many environmental factors occur independently of climate change and may be influenced by other socio-political factors, often overlooked in environmental migration studies. For example, changes in land use, urbanisation, overexploitation of natural resources, environmental pollution and geophysical natural hazards may each be key determinants of migration. Such environmental changes often have strong feedback loops—for example, rural to urban migration has significant repercussions on environmental degradation, air and water pollution, energy consumption and greenhouse gas emissions [70].

Environmental drivers have been found to be critical in many development studies. The Environmental Kuznets Curve (“EKC”) hypothesis purports that the early stages of economic development are coupled to environmental degradation and has been found to be true in many contexts [71]. In the context of urbanisation led by adaptive migration, this hypothesis suggests that urbanisation will result in further environmental degradation, with significant implications for future development [33,72], health [73], political security [74], and internal migration [75].

## 4. Applying the Model to the Case Study of Malawi

We now provide a brief example of applying the conceptual model to a case study. We select rural Malawi as a pertinent example of a climate-vulnerable society. Malawi is a land-locked country in southern Africa whose main economy is small-scale, rainfed agriculture, employing approximately 85% of Malawians [76]. As such, many people’s livelihoods as well as key source of food is highly climate sensitive. Already Malawi has witnessed an annual mean temperature increase of 0.9 °C since the 1960s, and whilst local rainfall patterns are difficult to accurately model, there has been an observed increase in frequency and magnitude of drought and flood events [77].

By conducting an in-depth literature review of Malawi’s political, demographic, environmental, social and economic makeup and then applying the conceptual approach described above by considering the impacts of climate change (primary, secondary and tertiary) to each key factor, we arrive at the case-specific model shown in Figure 2 below. 

A key advancement of this Malawi-specific model is that each variable is quantifiable using observational datasets. As such, it demonstrates how the application of the generalised conceptual model in Figure 1, to a local context, allows the creation of an astute, practical and measurable model, from which well grounded, policy-relevant research questions may be formulated and tested. By applying this methodological process, the Malawi-specific model that is generated is based on well-grounded assumptions and it holistically captures key variables that may be of relevance for future testing. Additional information about each variable can be found in Appendix A. Based on this conceptual model, the next step in the method would be to identify appropriate study and modelling techniques such as epidemiologic, mathematical, or integrated models to quantify the extent of each relationship depicted by the arrows in Figure 2. The insights from such models may therefore make possible evidence provision which can be particularly relevant for national adaptation, economic development and public health plans.

## 5. Complexities within the Model

Complexities naturally arise when taking an upstream, systems-thinking approach to migration determinants. There are two key complexities identified. Firstly, the acknowledgement of multilevel and interactions and feedbacks between drivers. Secondly the dynamicity of drivers and their connections over time and space. Despite these complexities, models must be transparent and provide results from which simplicity may be derived in order to be useful for decision makers and intervention planning.

To aid reflection upon such interactions, Figure 3 depicts a simple representation of the interactions between individual and classes of drivers. Each class of driver is represented by a funnel, from which a combination of both macro and micro drivers is filtered from an interconnected reservoir where drivers from different classes interact on a range of temporal, spatial and social scales. The combination of drivers at the individual migrant level results in a unique vulnerability profile and context which determines the migration decision made by each potential migrant. Climate change is again presented as an externality, cross-cutting all other driver classes and acting across the temporal and societal levels. As in Figure 1, each driver may vary over both time and spatial dimensions. However, modelling such dynamicity requires simultaneous understanding of drivers, their interactions, and their evolution through time and space. The insight that such dynamic modelling would allow may enable the effective identification of suitable intervention nodes for public health, land use and immigration policy to name but a few.

The concept of a vulnerability profile allows for the acknowledgement that each migrant has a unique set of drivers due to the multilevel and multitemporal combination of factors he or she is subjected to. In this way vulnerability may be conceived as a meta-driver of migration. The concept of vulnerability describes the ability of an individual or community to withstand and recover from a risk such as a disaster event [20]. Other meta-drivers include resilience and adaptive capacity [78,79,80]. Whilst vulnerability is a commonly used meta-driver in much climate migration literature, resilience is often the currency of choice in the fields of disaster management and climate change adaptation [81]. However, these terms are broad and often overlap and are even used interchangeably, rendering their distinction and usefulness within scientific analysis questionable. Despite this, such meta-drivers are the dialogue of choice for policy makers and must be utilised for research to have political relevance. However, care should be taken when referring to such meta-drivers and the contributing drivers as explored above must be contextually relevant and carefully selected.

## 6. Modelling Opportunities and Challenges

Previous conceptual models explore the linkages between climate change and migration with different assumptions and perspectives. The 2011 Foresight report identifies five key families of drivers and concludes that migration may be an adaptive strategy in the face of climate change and represents possibly the best to-date, globally accepted conceptual model for climate migration [10]. The report disputes the long-time argument that migration represents a failure to adapt in situ. This conclusion, however, fails to consider several key aspects of migration: firstly, the agency and social well-being of migrants involved at each stage of the migration process (prior to movement, in transit, and at host destination). Secondly, the level of agency afforded to would-be migrants during the migration decision—even as a supposedly proactive adaptation measure. Finally, the delicate line between forced and voluntary movement, based upon a composition of drivers and the bias of the person(s) awarding the classification.

The ongoing Lancet Commission on Climate Change and Health also presents an interesting framework where migration as a result of climate change is appropriately framed as a health challenge, and a public health opportunity [7]. This framework, however, does not give a large amount of consideration to intermediate drivers and various pathways by which climate change may drive migration or produce trapped communities. Helping to close this gap, and drawing on a range of political and economic, as well as health literature, the model presented by Sellers, Ebi and Hess considers a puzzle of immediate and longer-term drivers of social instability, with both climate shocks and migration as contributing factors and possible outcomes [82]. McMichael et al. [5] also present a foundational model whereby the basic links between climate change and migration are presented though driver interactions and dynamics are not discussed in depth.

Whilst this and other conceptual models encourage an upstream approach to environmentally induced migration, putting such thinking into practise presents further challenges. The paucity of empirical studies limits our understanding of how global climate change may threaten development and public health, particularly regarding the indirect impacts of climate change. Lack of suitable data and quantitative metrics needed to conduct such studies remains a perennial challenge. It is essential that these challenges be overcome through future data collection and empirical modelling.

Migration datasets are largely based on cross-sectional survey and census data whilst information about health and well-being, disaggregated by migration status, is largely lacking. Furthermore, collecting and disseminating such data present significant ethical and privacy concerns. For many drivers, proxies may be used. For instance, the Normalised Difference Vegetation Index (NDVI) may act as a proxy for natural resource availability [67]. Henderson et al. use a simple count of manufacturing industries as a proxy for urban industrial capacity when analysing the relationship between climate change and urbanisation in an African context [83]. Lu et al. [84] suggest the possibility of using out-migration rates as a proxy for changes in habitability. Neumann and Hilderink [17] present a range of possible datasets such as GLASOD for soil degradation and LADA for biomass production of earth observation land degradation data. However, each of these datasets has its own challenges concerning spatial and temporal resolution, uncertainty and effectiveness as a proxy. Furthermore, misalignment of datasets at the spatial, temporal and social levels creates further challenges in appropriately modelling migration determinants. Other, more squidgy drivers such as perceived political stability and social networks remain elusive to measurement and under-represented in quantitative studies.

The availability and quality of data in turn create methodological challenges for empiricists. Some studies utilise a range of statistical and epidemiological methods. However, traditional epidemiological methods each have their short-comings. Cross-sectional analyses do not allow for the temporal nature of drivers. Timeseries analyses are often impeded due to lack of sufficient data and the ability to control interactions between drivers across a range of temporal and spatial resolutions. Gravity models can capture linear push and pull factors at the macro level, though may struggle with ecological fallacy and in modelling of the more nuanced relationship between driver and migration outcome. Recent developments in mathematical models offer useful insight. Such models include improved agent-based modelling (ABMs) and multiagent systems approaches [21,52,85,86]. Study approaches must be chosen appropriately based on the assumed relevant determinants and their interactions, as choice of methods may have significant impact on the study results. The advantage of such systems approaches is that driver dynamics and interactions may be inbuilt and allowed to alter in timesteps. The individual nature of human decisions may also be captured through ABMs. However, ABMs require high-resolution data and are generally only applicable for small geographical scales. Economic approaches such as economic bargaining theory can also be used to explain some micro-level migration decisions such as the ‘healthy-migrant’ effect, whereby young and fit-for-work individuals may be more likely to move in search of work and remittance opportunities [87]. However, since climate change exists only as a macro factor, such micro-level considerations within current models of climate migration are often lacking.

Other approaches have been proposed to deal with complexity and dynamicity. Barbieri et al. [88] use a combined economic–demographic–climate model to understand the interactions between different classes of drivers over time (using appropriate proxies) in the northeast region of Brazil. Another emerging method is the use of Shared Socioeconomic Pathways (SSPs) to provide a combined set of scenarios for future population, urbanisation and wealth factors [89]. The SSPs are designed to be used in conjunction with climate change representative concentration pathways (RCPs) for future radiative forcing emission scenarios. Application of this approach can be seen within the 2018 Groundswell report who combine the RCPs and SSPs into three scenarios and use gravity modelling to provide a view of internal migration for three global regions [75].

Finally, we make a crucial note regarding the overall approach by scientists towards climate migration. Care must be taken when treading the literature of various typologies and terminologies which are necessarily subjective and vary by author and by discipline. Furthermore, climate migration may be studied through a variety of academic lenses. As such, the impact of different epistemologies on conclusions is complex and often overlooked [24]. Politically impactful research should attempt to transcend traditional research boundaries and avoid tribalism in science [90,91]. Indeed, in the pursuit of improved global health, research of climate migration should be contextually relevant, and politically pertinent and timely. One way to help achieve this is to adopt a pan-disciplinary approach such as the one demonstrated within this paper. Table 4 elucidates this point by demonstrating a selection of fields which contribute to the study of climate migration as an aspect of global health.

## 7. Conclusions

Ultimately, climate change may have critical impacts upon future migration across the globe and has significant implications for public health, human security and sustainable development. Climate change is already and will continue over the coming decades to contribute to large numbers of displaced persons [93], refugees [94], internal migrants [75], international migrants [26] and immobile and trapped persons [27]. As such, better understanding of the relationship between climate change and migration is essential for effective future policy planning in all sectors. This can be achieved through the systematic and homogenous application of robust conceptual frameworks to local contexts. A lack of data, particularly for low-income and indigenous settings, is a key set back which obstructs furthering our understanding. It also hinders the ongoing desire of academia, national and international policy makers to identify who are the climate migrants of today and of the future and count how many there are.

This paper has attempted to demonstrate the need for a flexible and pan-disciplinary approach to environmentally induced migration. Research which cross-cuts traditional discipline boundaries, in accordance with the planetary health viewpoint, is encouraged when using such a conceptual framework in the study of climate migration [95]. In this way, traditional pitfalls may be avoided, better reconciliation of macro and micro determinants may be achieved and more visibility of the dynamics of drivers and hence a more accurate understanding of their role in driving migration may be elucidated. However, the review within this paper is non-exhaustive and draws lightly on a wide range of literature and academic standpoints. As such, it is designed to demonstrate a nuanced approach to climate migration theory and application, rather than present a comprehensive “how-to” guide. Finally, it was beyond the scope of this paper to fully apply our conceptual model to mathematical and epidemiological quantitative models, providing instead a simple overview, though this is the natural progression of the research.

## Figures and Tables

**Figure 1 ijerph-17-06036-f001:**
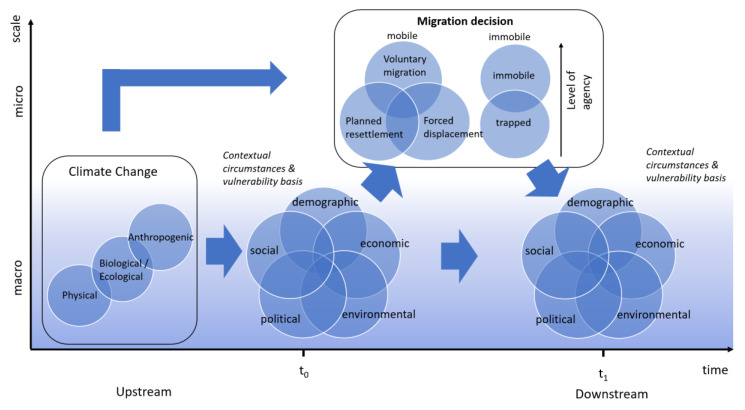
A conceptualisation of the pathways through which climate change may impact upon a system of vulnerability determinants that influence an individual’s migration decision.

**Figure 2 ijerph-17-06036-f002:**
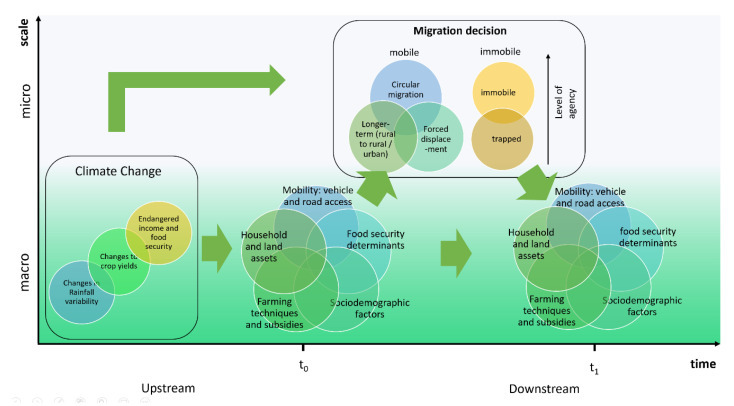
A conceptual model exploring the relationship between climate change and migration in the context of Malawi.

**Figure 3 ijerph-17-06036-f003:**
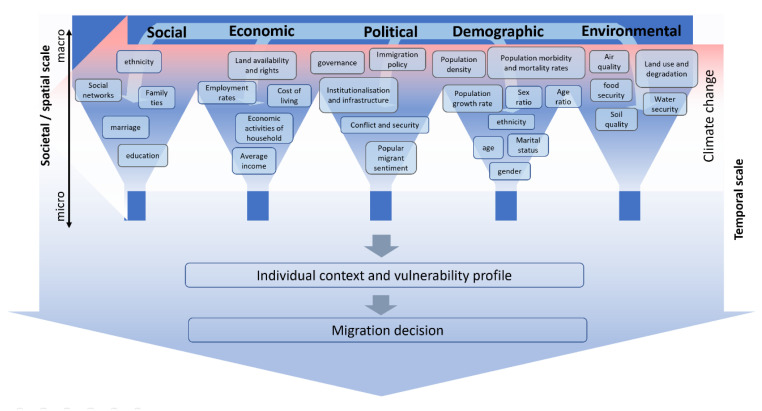
A simple representation of interactions between drivers and classes of drivers and the role of climate change as a source of external pressure on all drivers simultaneously.

**Table 1 ijerph-17-06036-t001:** Summary of the five categories of migration and their general qualities defined in terms of societal, temporal, spatial and agency levels.

	Type of Migration	Societal Level	Temporal Level	Spatial Level	Agency Level
1	Forced Displacement	Macro	Short term	Short distance	Low
2	Migration as an adaptive response	Micro	Varied	Varied	Varied
3	Planned resettlement	Macro	Permanent	Short distance	Low
4	Trapped	Micro	Varied	n/a	Low
5	Immobile	Micro	Varied	n/a	Medium/High

**Table 2 ijerph-17-06036-t002:** Key climate change characteristics that increase population vulnerability to environmentally induced migration.

Determinant	Temporal Scale
**Physical aspects**	
Changes in extreme or annual mean rainfall, resulting in a range of effects including droughts and floods	multilevel
Increased extreme weather events	fast
Land (including coastal) erosion	slow
Sea level rise	slow
Changes in average temperatures and temperature extremes	slow
**Biological/ecological aspects**	
Desertification	slow
Deforestation	slow
Soil degradation	multilevel
Changes to freshwater ecosystems including fish and other aquatic populations	slow
Changes to marine ecosystems including fish and other aquatic populations	slow
Changes to terrestrial ecosystems including changes to flora and fauna and vector-borne disease spread	multilevel
**Anthropogenic aspects**	
Changes in crop yield and agricultural productivity	multilevel
Changes in fishing catch	slow
Changes in water availability and security	multilevel

**Table 3 ijerph-17-06036-t003:** Non-climatic drivers of migration. Drivers are split into five classes: social, economic, political, demographic and environmental. Societal level refers to the societal scale at which drivers typically impact. Some drivers may exist both as micro and macro factors. Temporal scale refers to the typical timescale of change in each driver. Whilst there is no set demarcations, slow change refers to a change typically over years or decades and change fast refers to changes which may occur immediately or over a short timeframe of months. Static implies that factor is not usually time varying.

Determinants of Migration	Societal Scale	Temporal Scale
**Social**		
Family and societal relations and expectations	micro	slow
Migration and social networks (including remittance networks)	micro	slow
Changes in marital status	micro	slow
Education level	micro	slow or static
Ethnicity	multilevel	static
**Economic**		
Average household income	micro	multilevel
Key economic activity of household	micro	multilevel
Cost of living (e.g., consumer prices relative to income)	multilevel	multilevel
Employment rates and opportunities	multilevel	multilevel
Land availability and rights	macro	slow
**Political**		
Level of institutionalisation and infrastructure (governmental and other)	multilevel	slow
Conflict/security	multilevel	multilevel
Governance: policy incentives and state support initiatives and effectiveness of implementation and decision making by government	macro	slow
Specific migration policy and cultural sentiment towards migrants	macro	multilevel
**Demographic**		
Gender	micro	static
Marital status	micro	slow
Age	micro	slow
Ethnicity	multilevel	static
Sex ratio	macro	static
Population density	macro	slow
Population growth rate	macro	slow
Age ratio	macro	slow
Population morbidity and mortality	macro	multilevel
**Environmental**		
Soil quality	macro	slow
Land use/quality and degradation	macro	slow
Air quality	macro	Slow
Food security	macro	multilevel
Water security	macro	multilevel

**Table 4 ijerph-17-06036-t004:** An overview of the range of scientific disciplines which contribute to the study of climate migration, its drivers and impacts.

Discipline	Description
Human geography	Offers a range of frameworks and tools for studying human migration and its drivers.
Anthropology	Through the study of human behaviour, anthropological methods offer a deeper insight into the decision-making process behind migration, as well as the impacts of migration upon individual and societal well-being.
Ethnography	Ethnography offers a unique and rich insight into people’s opinions and decision making.
Political sciences	Political sciences may be used to explore the effects of policy on immigration, as well as the geographic, economic and social drivers of migration policy and sentiment.
Economics	Both macro and micro economics can be used to quantify migration as well as study the economic drivers and impacts of migration. For example, in the case study of Malawi, econometric modelling could be applied in the study of the impact of failed crops on household wealth and as such on migration.
Mathematics	A range of mathematical models are used in the study of migration such as system dynamic models, agent-based models, gravity models, and diffusion models.
Epidemiology	Environmental epidemiology can be used in the study of migration and its drivers. For example, the field of ecological public health supports the exploration of the relationships between the biological and material realms [92].
Disaster risk reduction sciences	Disaster risk reduction relates mainly to sudden-onset events and short-term, forced displacement and as such provides cross-over to the field of migration science.
Environmental sciences/agricultural sciences/hydrology/climate modelling	These are examples of fields which are relevant to the drivers of migration. For example, in the case study of Malawi, agricultural sciences may help model the impacts of climate change on crop yield.
Computer sciences	Computer science is used in migration studies to model and simulate migration and its quantifiable drivers and impact.
Sociologists	Sociology can be used to study migration and its impacts at the societal level, with special interest in demographic makeup and the social structure of migrant (and non-migrant) communities.
Demography	The study of population dynamics and structure places migration as a core component.

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
