# Peer review of "A Critical Analysis of the Drivers of Human Migration Patterns in the Presence of Climate Change: A New Conceptual Model"

_ijerph, 2020, doi:10.3390/ijerph17176036_

Round 1

Reviewer 1 Report

  • Line 178 “Error! Reference source not found.” Please delete this sentence.
  • Lines 178-179. “summarises these classifications according to the key societal, temporal, spatial, and agency markers.” This sentence is not correctly written. Please check it.
  • 1. Please enlarge the font of descriptions in the picture.
  • 2. Please enlarge the font of descriptions in the picture.
  • Line 422. The sentence should start with a capital letter.
  • 3. Please enlarge the font of descriptions in the picture.

Author Response

Thank you so much for your comments. Please find our responses within the attached.

Reviewer 2 Report

I considered this article to be of very high quality.  It presents a good foundation for further development of a model to predict human migration patterns.  I have very few edits to recommend and pose a few questions that the authors may or may not have considered.

Line 126:  How early does, or should such intervention design begin?

Line 156: Do you think that international movements could describe all long-distance movements?  Malawi is a relatively small country as compared to say Brazil.  Is it feasible that international movements between small countries would constitute something on a smaller scale?

Line 178: Check reference.

Line 213:  In regards to a host area, what considerations should be given to a host area’s willingness to accept refugees?  Could an influx of refugees result in a hardening of international borders leading to further instability for the refugees and for the potential host country?

Line 230: In regards to host country interventions, have you considered the benefits to developed countries in helping mitigate climate damage in less developed counties as a way to attenuate migration.

Line 288: Check reference.

Lines 319-321:  Is it possible that water conflicts could affect even affluent countries?  I wonder what could happen should the Ogallala Aquifer in the US were to prove insufficient to water the mid-American bread belt.  Another example might be water rights conflicts in California or the water requirements of nut orchards vs staple food crops.

Line 338:  This section again reminds me of concerns about hardening of international borders.  Recent examples might include rising nationalism resulting in Brexit or Trumpism.

Line 366: In this section, is it possible that aging populations in developed countries could benefit from a younger population of migrants from afflicted countries?  How could the mindset of the host country be changed to understand this possible benefit in the face of nationalism mentioned above?

Line 435:  What new complexities might arise as migration begins?  There are certainly many unknowns, but there will just as certainly be a cascade of consequences.

Line 529:  I appreciate the multi-faceted approach you have taken to examining this problem.  Did you, by chance, consider the work of the labor economist Kreuger?

Author Response

(The authors gave the same response as above.)

Reviewer 3 Report

The article talks about the impact of climate change on migration, which today is a very important issue. However, some aspects could be improved:
1. In the introduction, I suggest expanding the importance and contribution of research in the literature. Knowing the impact of climate change on migration, how could these results improve the well-being of the population? How would these results help to plan the application of the policies?
2. It would be desirable to transfer this conceptual model to a mathematical model.
3. Authors must review the cited in the text. Throughout the paper there are citation errors.
4. Authors should clearly establish the limitations of the conceptual model.
5. Authors should also explain the importance of data collection and empirical modeling in future research.

Author Response

(The authors gave the same response as above.)
